# Chronic Pain in the Elderly: Mechanisms and Distinctive Features

**DOI:** 10.3390/biom11081256

**Published:** 2021-08-23

**Authors:** Andrea Tinnirello, Silvia Mazzoleni, Carola Santi

**Affiliations:** 1Anesthesiology and Pain Medicine Department, ASST Franciacorta, Ospedale di Iseo, 25049 Iseo, Italy; 2Second Division of Anesthesiology, Intensive Care & Emergency Medicine, University of Brescia at Spedali Civili Hospital, Piazzale Spedali Civili 1, 25100 Brescia, Italy; silvia.mazzoleni09@gmail.com (S.M.); carola.g.santi@gmail.com (C.S.)

**Keywords:** chronic pain, opioids, aging, sensitization

## Abstract

Background: Chronic pain is a major issue affecting more than 50% of the older population and up to 80% of nursing homes residents. Research on pain in the elderly focuses mainly on the development of clinical tools to assess pain in patients with dementia and cognitive impairment or on the efficacy and tolerability of medications. In this review, we searched for evidence of specific pain mechanisms or modifications in pain signals processing either at the cellular level or in the central nervous system. Methods: Narrative review. Results: Investigation on pain sensitivity led to conflicting results, with some studies indicating a modest decrease in age-related pain sensitivity, while other researchers found a reduced pain threshold for pressure stimuli. Areas of the brain involved in pain perception and analgesia are susceptible to pathological changes such as gliosis and neuronal death and the effectiveness of descending pain inhibitory mechanisms, particularly their endogenous opioid component, also appears to deteriorate with advancing age. Hyperalgesia is more common at older age and recovery from peripheral nerve injury appears to be delayed. In addition, peripheral nociceptors may contribute minimally to pain sensation at either acute or chronic time points in aged populations. Conclusions: Elderly subjects appear to be more susceptible to prolonged pain development, and medications acting on peripheral sensitization are less efficient. Pathologic changes in the central nervous system are responsible for different pain processing and response to treatment. Specific guidelines focusing on specific pathophysiological changes in the elderly are needed to ensure adequate treatment of chronic pain conditions.

## 1. Introduction

The International Association for the Study of Pain (IASP) defines pain as an unpleasant sensory and emotional experience associated with or resembling that associated with actual or potential tissue damage [1]. Pain is defined as chronic when it lasts or recurs for longer than 3 months [2].

As the population of developed countries ages, there has been an increase in the prevalence of conditions associated with persistent pain across settings of care [3,4,5].

Survey-based studies in Europe show that chronic pain incidence increases with age, with an estimated prevalence ranging from 38 to 60% in people over 65 years old [4,5]. Patients commonly report osteoarthritic back pain, especially in the low back or neck (around 65% of all chronic pain patients), musculoskeletal pain (around 40%), peripheral neuropathic pain (typically due to diabetes or postherpetic neuralgia, 35%), and chronic joint pain (15–25%) [3,6]. Despite the high prevalence of painful conditions in the elderly, specific guidelines regarding pain management in this population have not been developed, and only general advice regarding pharmacological treatment and clinical assessment have been proposed.

Pain management in the elderly can be extremely challenging for various reasons: cognitive impairment makes pain assessment difficult since patients may not be able to precisely locate the pain and describe its intensity and features, pharmacological interactions are frequent in patients under multiple medications, and pharmacodynamic and pharmacokinetic changes in drug metabolism make medication responsiveness unpredictable [7,8,9]. Regarding pharmacological changes in the elderly, interindividual variability in opioid sensitivity has been reported in several studies with reports of increased, decreased, or unchanged sensitivity in older patients compared to younger adults [10,11,12,13]. However, general advice when starting an opioid treatment in elderly patients is to use lower dosages [13]. Extensive research has been done on pain assessment tools in patients with dementia or cognitive impairment, but only a minority of studies investigated the uniqueness of pain in the elderly from a physiological point of view [14]. In fact, experimental data on age-related changes in pain perception are scarce and contradictory [15,16,17].

In a complex and delicate setting as pain management in frail and elderly patients, the knowledge of specific mechanisms could be crucial for improving the quality and efficacy of treatments. Pain processing is extremely complex and consists of the transduction and transmission of a mechanical, chemical, or thermal noxious input from the peripheral receptor to the brain. Many anatomical structures are involved in this process: myelinated and unmyelinated fibers, dorsal root ganglia, spinal cord, supraspinal structures [18]; all these stations can be involved in age-associated changes that can lead towards an altered pain processing. According to a recent systematic review and meta-analysis, including 40 studies on 6955 patients older than 60 years, age-related changes appeared to gain physiological substance when heat stimulation was used as a physical stressor, rather than pressure and electrical current [19]. The application of heat stimuli emerged to be especially appropriate to demonstrate an age-related decline in pain sensitivity. Heat sensation is mediated by C-fibers, which are located mainly in the superficial tissue; pressure pain instead is mediated by deep tissue nociceptors with an electrical current that activates Aδ-fibers directly, bypassing any receptor mechanism [19]. Earlier narrative reviews suggest a tendency toward an increase in pain thresholds and decrease in tolerance thresholds with increasing age, however, these findings have not been confirmed [19,20,21]. The emerging consensus is that the neurophysiological response to tissue injury and nerve damage changes with age [22]. Mechanisms such as hyperalgesia and temporal summation (a phenomenon in which repeated and equal-intensity noxious stimuli at a specific frequency cause an increase in the pain experienced) are hypothesized to be more common in older people [23,24].

In this review, we focused on pathophysiological changes in pain transmission and processing to underline which mechanisms are predominant in the elderly and why pharmacological treatments are often inadequate. We divided the evidence into three major topics according to the station of pain processing: peripheral, spinal, and supraspinal, and we particularly highlighted the role of descending inhibitory mechanisms such as the opioid system, a crucial regulator in pain processing; as previously mentioned, opioid sensitivity presents interindividual variability in elderly, we tried to find a molecular basis for this phenomenon. The majority of data presented in this review are derived from animal models where pain was induced by different mediators (e.g., heat, pressure, chemicals), even if these experimental models may not fully reflect the human mechanisms, we noted in the next where human models were used.

## 2. Materials and Methods

Narrative review. The literature search was conducted using the PubMed, MEDLINE/OVID, and SCOPUS databases with papers published until May 2021. 

## 3. Results

### 3.1. Peripheral Nerves and Receptors

The change of the peripheral nervous system in aged subjects is evident, however, the responsible mechanisms are very controversial and often unknown [25,26]. Weyer et al., comparing C-fibers in healthy mice, found that the action potential firing response to mechanical stimuli was higher in aged mice than in the younger ones, thus resulting in higher sensitivity in the aged group [25]. During acute inflammation, a significant sensitization in action potential firing in C-fiber afferents in young animals is noted; whilst this is not evident in the old ones [25]. This may be explained by a higher baseline inflammation level in the older subjects, which would explain the higher baseline sensitivity as well [25]. Consistently, during chronic inflammation, C-fibers of young animals exhibit a greater reduction in firing rates than old animals. Strikingly, in young animals during chronic inflammation, there is a significant elevation in the number of C-fiber afferents displaying spontaneous firing. Similar trends are observed in C-fiber responsiveness if the stimulus is either mechanical or induced by the interaction of capsaicin with transient receptor potential vanilloid 1 (TRPV1) [25]. The mechanisms of peripheral sensitization through aging do not seem to depend on gene expression of ion channels but rather by the sensitization of the mechanoceptors [25]. The only gene found to be expressed at higher levels in aged animals is TRPV1, but it seems to play a major role in the synapse in the dorsal horn of the spinal cord rather than on the skin [25]. Previously, Taguchi et al. had hypothesized that the decreased expression of TRPV1 channels in cutaneous nerves might be involved in the decrease of heat-responsive fibers [27]. 

Chakour et al., in a study on healthy volunteers, demonstrated the Aδ- fibers impairment in older adults [28]. Using the compression of the superficial radial nerve at the wrist, they blocked the Aδ-fibers, allowing C-fibers alone to conduct cutaneous stimuli in two groups of young (20–40 years) and old (over 65 years) adults. In the two groups, there was a significant age-related difference in pain and thermal thresholds when both Aδ and C-fibers were intact, while the difference disappeared when the C-fiber information alone was conducted. Aδ-fibers transmit sharp, well-localized pain; their selective impairment would explain the altered epicritic pain perception in older adults [25,28]. 

Experimental research in both animal and human models has highlighted a loss of myelinated fibers and a decrease of myelin in aged individuals [26,29,30]. This leads to a reduction of the axonal size and an increasing irregularity of the shape, which has been related to a decrease in neurofilament mRNA expression with aging. Consequences of this are wide incisures in the myelin, segmental demyelination, swollen demyelinated axons, and collagen pockets. The decrease of mRNA expression involves some major structural proteins of the peripheral nervous system: protein 0 (P0, involved in the maintenance of the multilamellar structure), peripheral myelin protein 22 (PMP22, crucial for the formation of myelin), myelin basic protein (MBP, located in the cytoplasm of myelin lamellae, involved in myelin compaction), myelin-associated glycoprotein (MAG), and connexin 32 (Cx32). The endoneurial ATP and creatine phosphate levels decline with age, as well as the nerve blood flow, demonstrating the reduced energy requirements and energy stores [26]. The rate of axonal CGRP (calcitonin gene-related peptide), which regulates the metabolism and function of muscle acetylcholine receptors, declines with age, contributing to changes in junctional acetylcholine receptors [26].

Procacci et al. in 1970 justified the altered thermal and pain sensitivity in the elderly with the difference in skin thickness, but further studies contested this hypothesis [31]. According to a recent review, age differences appear more evident for non-contact stimuli (laser, radiant) than for contact ones. This might be explained by the compression of the epidermal junction, which would minimize the influence of age-related changes in the properties of the skin [11]. Biochemical studies have documented a marked reduction in substance P content in aged human skin, other than in the lumbar dorsal root ganglion cells, and reduction of levels of CGRP in the mesentery of aged rats [11,26].

According to Taguchi et al., there is a significantly decreased number of mechano- and heat-responsive C-nociceptors (CM) and an increased number of mechano- and thermo-insensitive C-fibers (CMi) in aged mice [27]. The relative receptors have then been sorted into subclasses by mechanical- and thermo-responsiveness to stimuli: in the aged group, there is a remarkable increased number of C-mechanical nociceptors, whilst the heat-responsive receptors are reduced [27]. Another study found a consistent change in the ratio of the CM and the CMi; fewer polymodal nociceptors with a mechanical response are found in aging subjects in comparison to young subjects [32]. Degeneration of the fibers seems to be the leading mechanism, but in the aged, sensitization and spontaneously activated C-fibers are also seen [32]. The link may be a surplus in nerve growth factor (NGF); NGF present in the skin is not taken up by degenerated fibers and it may be responsible for the sensitization of CMi to mechanical and thermal stimuli, causing spontaneous activity and lowered heat thresholds [32]. The mechano-responsive C-fibers of aged subjects also show more slowing to high-frequency stimulation and it has been demonstrated that the sodium channel availability is related to conduction velocity of unmyelinated fibers [33]; the loss of sodium channels with age may be due to metabolic changes and reduction in ATP supply [32]. Age-related changes in peripheral afferents do not seem uniform in the different organs, for example, the mechanical response seems to be emphasized in joint and muscular fibers. This might be explained by changes in properties and structures of the tissues with age (elasticity, initial tension, thickness, collagen composition) [27]. Conduction velocity of C-fibers is decreased in the mechano-responsive C-nociceptors in the aged; the two possible mechanisms involved are the hyperpolarization by the axonal sodium–potassium–ATPase and the number of available sodium channels in aged unmyelinated fibers [27], these findings have been questioned by Weyer et al. who demonstrated that conduction velocities in C-fibers are unaffected either by age or baseline nerve injury [25].

Impairment of the peripheral nervous system with age has been evidenced in the myenteric plexus as well. A decreased pain perception in the esophagus has been demonstrated by blowing a balloon 2 mL at a time in young and old adults and measuring the pain threshold [34]. Previously, in agreement with this, Meciano noted a decrease in density of neurons (a decrease in neuron number per square centimeter by 22 to 62%) in all four different segments examined of the esophagus wall in aged persons [35].

Despite the paucity of evidence and the conflicting results, these findings indicate that peripheral nociceptors have little influence on the pain sensation in aged individuals [25]. Moreover, mechanical, rather than heat, stimuli are more likely to produce pain due to the relative reduction in heat-activated receptors and Aδ fibers [11,27,28].

### 3.2. Spinal Cord and Descending Modulating Systems

Older adults have shown widespread degenerative changes in spinal dorsal horn sensory neurons, such as a marked loss of myelin, evidence of axonal involution, particularly in the medial lemniscal pathways, and altered spinal neurochemistry [36]. Immunohistochemical studies reveal decreased labeling of CGRP, substance P, and somatostatin in the cervical, thoracic, and lumbar dorsal horn of aged rats [37,38]. The descending modulation displays age-related impairment in opioid and non-opioid mechanisms; there is strong evidence of a progressive age-related loss of serotonergic and noradrenergic neurons in the dorsal horn, especially in Lamina I of the spinal dorsal horn [11,36,39,40]. The extent of this alteration is quite large, with elderly people showing less than a third of the strength of the induced endogenous inhibitory effects on pain sensitivity compared with young adults [41,42,43]. 

Studies on temporal summation of heat pain support the hypothesis that endogenous pain inhibition appear to be reduced in the elderly [7,44]. Inability to modulate painful processes contributes to increased vulnerability in the development of chronic pain after injury or illness [45]. A psychophysical study used a capsaicin-induced model of temporary injury to investigate primary and secondary hyperalgesia in young and old adults [46]. Both groups displayed a similar magnitude of pain, hyperalgesia, and flare in response to the topical application of a 5 mg/mL dose of capsaicin. However, older adults exhibited a delay in the report of capsaicin-induced pain and a markedly slower resolution of the area of punctate hyperalgesia [46]. The state of mechanical hyperalgesia in the unstimulated skin surrounding the primary zone has been shown to be mediated by sensitized spinothalamic tract neurons, such as wide dynamic range (WDR) and high threshold (HT) neurons [47]. Those findings suggest a possible age-related change in spinal cord sensitization processes. A reduced capacity of the aged CNS to reverse the sensitization process might be reflected by a slower resolution of punctate hyperalgesia. Translated into the clinical situation, it would suggest a slower recovery and an increased duration of post-injury tenderness in adults of advanced age [46].

Recent research has highlighted the role of glial cells in chronic pain processing; these cells intervene by releasing pro-inflammatory cytokines, chemokines, and proteases that can amplify peripheral pain signals at the spinal level [24,45,48]. Microglia interacts with spinal neurons at the site of injury or disease, as well as remotely, responding also to pro-inflammatory signals released from peripheral cells of immune origin, including mast cells [49]. Microglia activation and neuroinflammation are the main topics for researchers; these mechanisms are involved in the transition from acute to chronic pain, particularly in neuropathic pain. Neuroinflammation is among the age-associated changes that occur in the central nervous system; it has been suggested that aging is linked with chronic innate immune activation and significant changes in monocyte functions, which may have implications for increased low-grade chronic inflammation and for the development of age-related diseases [50]. Thus, macrophage activation, together with inflammatory monocytes, contribute to the subclinical chronic inflammatory process dubbed “inflammaging” [51,52] Both mast cells and microglia, the main immune-resident cells in the CNS, undergo a change in reactivity with age. Mast cells are altered in older adults and their maturation decreases with age in various tissues, including the endoneurial compartment [53]. Aged mast cells show an increase both in their sensitivity to inflammatory mediators and state of degranulation [54]. In the spinal cord and thalamic pain nuclei of older adults, primed microglia are activated and resistant to regulation, facilitating the onset of chronic and/or neuropathic pain as well as a state of neuronal hyperexcitability (central sensitization) [45]. In animal models, it has been demonstrated that aging modulates functional changes in the microglia of the trigeminal spinal subnucleus caudalis and regions following palatal mucosal injury, resulting in an enhancement of mechanical allodynia due to the augmentation of neuronal hyperexcitability regulated by age-related microglial activation [55].

Other animal studies indicate that aging causes changes in the expression of a range of transcripts similar to those seen in neuropathic pain. C-X-C motif chemokine 13 (Cxcl13), for example, is highly upregulated following nerve injury, and is largely involved in inflammatory responses and glial function in spinal dorsal cord. Cxcl13 also shows a pronounced increase in gene expression with age alone, displaying critical changes in pain signaling in the elderly [56]. A decrease of myelin proteins has been reported to correlate with increased glial activation because of the production of pro-inflammatory cytokines that can compromise white matter integrity [56]. These results are consistent with observations relating to age-dependent changes of myelinated Aδ type fibers as well as unmyelinated C-type fibers [29,30].

### 3.3. Brain and Supraspinal Changes

The lateral pain system consists of spinothalamic tract neurons that ascend via the ventro-posterior lateral thalamus onto the primary and secondary somatosensory cortices (S1 and S2), which, in turn, code the location, intensity, and quality of the sensation [53]. Another major pathway branches off at the level of the medulla. It ascends via the medial thalamus to the hypothalamic nuclei and the limbic regions, which include the cingulate cortex, the insula cortex (IC), and the prefrontal areas. These areas are known to be involved in the control of emotion, arousal, and attention [57]. This medial pain pathway is proposed to mediate the unpleasant, affective dimensions of pain and the motivation to escape from the noxious event [57]. Various cortical and subcortical structures are involved in the cerebral response to nociceptive stimuli: the anterior cingulate cortex, the insular cortex, the thalamus, the primary and secondary somatosensory cortices [58]. Cerebellum, amygdala, basal ganglia, and midbrain are less commonly involved in the nociceptive process.

Neuronal death, loss of dendritic arborization, and neurofibrillary abnormalities occur throughout the aged cerebral cortex and to a lesser degree in the midbrain and the brainstem. The degeneration mostly involves brain structures associated with pain processing. Chronic pain creates specific alterations in the brain structure. The extent of structural reorganization follows distinct trajectories for different types of chronic pain [59]. A few neurotransmitter concentrations, such as GABA, serotonin, dopamine, noradrenaline, and glutamate decrease with age [11]. A reduction in the density of serotonin and glutamate receptors has been demonstrated in the cerebral prefrontal cortex. Since all these molecules are involved in pain processing and modulation, their decrease suggests that neurochemicals necessary for pain modulation may not be sufficiently available in older adults [60]. A reduction in GABA concentration has been correlated with pain intensity in patients with painful chronic osteoarthritis and fibromyalgia (with a compensatory increase in GABAa receptors). This suggests a disinhibition of brain regions, like the anterior cingulate cortex, as a common mechanism that increases pain severity in different kinds of pain [61,62]. The same reduced level of GABA has been found in elderly subjects with chronic pain [18,25]. A reduction in brain β-endorphin concentration could explain the reduced efficiency of the inhibitory systems [63]. A study on aged patients with chronic knee pain showed a positive correlation between circulating plasma β-endorphin level and higher pain sensitivity [64]. Elevated resting plasma β-endorphin (which does not reflect β-endorphin brain concentration) might be an indicator of reduced endogenous opioid analgesic capacity, probably due to the downregulation of the opioid receptors [64].

The expression of the opioid receptors in the central nervous system changes with age. The KOR level (which mediates aversive feelings), instead appears to remain unchanged with age [65]. On the contrary, a study on adult and older rats identified a marked age-related reduction in MOR expression in the periaqueductal grey matter, which corresponded to higher morphine requirements to obtain analgesia in older rats compared to adults [66].

With specific MRI neuroimaging studies, the brain “age” can be defined and can differ from the biological age of the subject [67]. Chronic pain is associated with an “older” brain relative to the individual’s chronological age [68]. This association was particularly evident in patients with chronic pain who had not received any treatment in the previous 3 months. In another study, however, patients who had received adequate medical treatment for their pain condition showed no differences in brain aging compared to matched controls [69,70]. In a functional MRI (fMRI) study on patients with early-stage Alzheimer’s disease, functional brain imaging data produced no evidence of reduced pain-related activity in the medial pain regions in patients compared with healthy controls, indicating that the affective/emotional aspects of the pain experience are not selectively diminished in these patients [71]. Central processing of painful stimuli appears to remain intact even when brain tissue is damaged. Other studies using fMRI in extreme ages (up to 97 years old) revealed that brain activation patterns in response to noxious stimuli remain unchanged [72]. Hippocampal volume tends to be smaller in older adults who report more severe pain (chronic and acute). Lower levels of N-acetylaspartate/creatine (a marker of neural integrity) have been found in the same subjects [73]. Another fMRI study in elderly patients with chronic disabling low back pain found decreased white matter integrity (despite a normal brain volume) in the splenium of the corpus callosum compared to healthy subjects [74]. These changes partially overlap with the findings in patients with late-onset depression, raising the question of whether chronic disabling pain can be a somatic manifestation of late-onset depression or if structural changes induced by chronic pain predispose patients to develop a depressed state [74].

## 4. Discussion

Research on the pathophysiological foundations of chronic pain in the elderly is limited and often conflicting. However, multiple pieces of evidence demonstrate that the age-related changes in peripheral and central nervous systems affect all levels of pain processing.

Decreased somatosensory perception can be related to the loss of noci- and mechanoreceptors and to the reduced blood flow to the skin. Neuronal fiber loss and reduced conduction velocity are also associated with reduced sensitivity. Since peripheral nociceptors contribute little to the development of chronic pain in the elderly, peripherally acting analgesics (such as NSAIDs) can have little effect in an aged population [25].

The diffuse inhibitory noxious controls appear to be less efficient in the elderly [42].

Thermal and electrical pain thresholds increase after repeated cold pressure stimulations, but the magnitude of the increase is significantly greater in younger subjects. These findings could indicate that, in the absence of efficient inhibition, repeated painful stimuli could produce a significant central sensitization. Moreover, older adults exhibit pain facilitation with a concurrent rather than an alternating noxious stimulus [75]. The diffuse reduction in descending pain inhibition activity, due to a reduced concentration of chemical mediators, exposes the elderly patients to the risk of developing disabling pain with repeating painful stimuli that are not counteracted by modulating systems. Glial cells activation in the spinal cord is involved in central sensitization, amplifying pain signals [24]. The link between glial cells activation, neuroinflammation, aging, and pain could lead towards important therapeutic considerations. A viable therapeutic strategy may be the target of non-neuronal cells through endogenous regulators of neuroinflammation. Palmitoylethanolamide (PEA) is an endogenous cannabinoid-like compound, part of the complex homeostatic system controlling the basal threshold of inflammation. It promotes the resolution of neuroinflammation and pain [76]. Animal studies demonstrated that prolonged PEA treatment has a direct peripheral effect reducing endoneurial edema, mast cell recruitment, and activation at the injury site, normalizing the sensitivity and function of primary somatosensory neurons [76]. It also influences the behavior of non-neuronal cells (microglia, mast cells, astrocytes) on spinal and supraspinal levels, thus reducing neuroinflammation [77,78].

At the brain level, despite a significant reduction in grey and white matter, typical of the aging process, fMRI studies revealed that the brain activation in response to painful stimuli and central processing pathways remain unchanged even in extreme age and in moderate cognitive impairment. Significant changes in brain structure have been demonstrated in elderly patients with chronic pain. It is unclear whether these changes are caused by chronic pain, or they can be considered a predisposing factor to develop more severe pain perception. An “older” brain is characterized by decreased somatosensory perception, deficient endogenous pain inhibition, lower positive affect, less agreeable personality, and being less emotionally stable [74].

While the density of MOR in supraspinal areas appears to be reduced by age (indicating higher morphine dosages required to obtain comparable analgesia with younger patients), comorbidities, concomitant medications, and changes in metabolic pathways could counteract this effect. We hypothesize that these factors could explain the difference in opioid sensitivity seen in clinical studies [69,70].

## 5. Conclusions

Pain management in the elderly remains a challenging issue for physicians. The pathophysiological changes associated with age involve structures related to pain processing. Even if pain processing appears to be efficient overall, in patients with dementia as well, multiple changes are responsible for an altered transmission and processing of painful stimuli. Table 1 resumes the age-associated modification we described in the text. Despite a reduced sensibility for painful stimuli (indicating that older individuals are less likely to report mild pain), age is associated with a marked reduction in the efficiency of inhibitory systems. Therefore, elderly patients are more prone to develop severe pain if stimuli are repeated (as in the case of osteoarthritis). Moreover, the brain changes associated with age can increase the emotional component usually associated with pain. Peripheral sensitization contributes marginally to the pain experience in advanced age.

A multimodal strategy involving psychological interventions, central acting medications, and interventional treatments should be used in elderly patients with chronic pain. Considering the large involvement of the microglia and mast-cells, palmitoylethanolamide may be considered as a co-treatment in the elderly, while peripherally acting drugs and interventions are probably less effective [45].

A limitation of this review is that most studies involved animal models of pain, where pain was induced by different mediators and stimuli, thus, whether these findings can be translated into human models is uncertain.

## Figures and Tables

**Table 1 biomolecules-11-01256-t001:** Pathophysiological features of pain in the elderly.

**Peripheral Nerves and Receptors**	Higher action potential firing response
	Lower sensitization of the potential firing in C-fibers during acute inflammation
	Lower expression of TRPV1 channels in cutaneous nerves
	Reduction in myelin and myelinated fibers, resulting in A-δ fibers impairment
	Decrease in mRNA expression of many proteins: P0, PMP22, MBP, MAG, Cx32
	Reduction in endoneurial ATP and creatine phosphate levels
	Reduction of substance P
	Increased ratio of the mechano- and heat-responsive C-nociceptors (CM) and the mechano- and thermo-insensitive C-fibers (CMi)
	Loss of sodium channels in unmyelinated fibers
	Decreased density of neurons in the myenteric plexus
**Spinal Cord** **and Descending Modulating Systems**	Decreased labeling of CGRP, substance P, and somatostatin in the spinal dorsal horn
	Loss of serotoninergic and noradrenergic neurons in the Lamina I of the dorsal horn
	Endogenous pain inhibition appeared to be reduced in the elderly
	Increased microglial activation and neuroinflammation
	Increased sensitivity of mast cells
	Increased degranulation of mast cells
	Increased in gene expression of chemokines, like Cxcl13
**Brain and Supraspinal Structures**	Decreased concentration of GABA, serotonin, dopamine, noradrenaline, and glutamate
	Reduction in the density of serotonin and glutamate receptors in the prefrontal cortex
	Reduced concentration of MOR
	Decreased hippocampal volume and levels of N-acetylaspartate/creatinine in adults who had experienced more pain
	Decreased white/grey matter in the splenium of the corpus callosum in adults who had experienced more pain

## Data Availability

No new data were created or analyzed in this study. Data sharing is not applicable to this article.

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
