# Peer review of "Chronic Pain in the Elderly: Mechanisms and Distinctive Features"

_biomolecules, 2021, doi:10.3390/biom11081256_

Round 1
Reviewer 1 Report
Dear Santi and colleagues
I would like to praise the manuscript, the idea of the work is excellent. The level of care you took to research and write was very good. But I would like to suggest a few points to make the work more attractive.
1. I think one figure in the conclusion would make the manuscript more attractive. As well as a concept map about pain in the introduction, given that you address in the results each point that the pain goes through until it reaches the central nervous system
2. In paragraph 53, where you have written about thermal and mechanical pain, I missed reporting on chemical pain in elderly people. Is there data on this subject? If so, you could add and write a little more and discuss this data.
3. The sentence starting at line 61 and going through 64 could be rewritten. It's a little confused.
4. What does temporal summation mean?
5. Are there no data on neuroplasticity in the elderly population suffering from chronic pain? If so, you could write a little.
6. In line 69, What are examples for cognitive impairment makes pain assessment difficult?
7. In the manuschrist, you highlighted the opioid system, so it should be clear in the objectives.
8. What is the role of A delta fiber in pain and its correlation with age?
9. Change ml to mL
10. On line 179, you write neuronal density. The unit for density is mass under volume. And it has the following writing "by 22 to 62% of neurons per square centimeter". Is this correct? Wouldn't it be better to write area instead of density?
11. In the spinal cord results, you do not write the change in the GABAergic pathway. Are there any changes with neurotransmitter GABA in elderly people?
12. Line 219, you wrote "mast cell density is", wouldn't it be better to write mast cells are? Remove density, what do you think?
13. Is there no correlation between chronic pain and patients suffering from neurodegenerative diseases? I think that writing was missing, in case there is any correlation.
Author Response
Dear reviewer, thank for your notes, we highlighted our answers in the list below.
- I think one figure in the conclusion would make the manuscript more attractive. As well as a concept map about pain in the introduction, given that you address in the results each point that the pain goes through until it reaches the central nervous system. We added a tab in the conclusions summarizing the main findings described in the text
2. In paragraph 53, where you have written about thermal and mechanical pain, I missed reporting on chemical pain in elderly people. Is there data on this subject? If so, you could add and write a little more and discuss this data.We haven’t found reports on chemical pain in the elderly.
3. The sentence starting at line 61 and going through 64 could be rewritten. It's a little confused. We rewrote the sentence
4. What does temporal summation mean? We added the definition
5. Are there no data on neuroplasticity in the elderly population suffering from chronic pain? If so, you could write a little. We expanded the section on spinal changes adding information on neuroinflammation and microglia activation, we found works on neuroplasticity and depression or cognitive impairment but not specifically on chronic pain in the elderly
6. In line 69, What are examples for cognitive impairment makes pain assessment difficult?We added an example of this
7. In the manuschrist, you highlighted the opioid system, so it should be clear in the objectives.We expanded the objectives section accordingly
8. What is the role of A delta fiber in pain and its correlation with age? Aδ-fibers transmit sharp and localized pain, we added this description in the text, their impairment in the elderly has been demonstrated by Chakour (ref.28)
9. Change ml to mL.We changed the unit.
10. On line 179, you write neuronal density. The unit for density is mass under volume. And it has the following writing "by 22 to 62% of neurons per square centimeter". Is this correct? Wouldn't it be better to write area instead of density? The authors of this paper investigated the neuron number per square centimeter as a density parameter, we rewrote the sentence accordingly.
11. In the spinal cord results, you do not write the change in the GABAergic pathway. Are there any changes with neurotransmitter GABA in elderly people? We wrote about GABA in the supraspinal section because the decreased concentration of these molecule has been investigated in these regions but the same conclusion could be taken for the spinal cord either.
12. Line 219, you wrote "mast cell density is", wouldn't it be better to write mast cells are? Remove density, what do you think?Yes it’s better, we changed the sentence.
13. Is there no correlation between chronic pain and patients suffering from neurodegenerative diseases? I think that writing was missing, in case there is any correlation.Yes there is a correlation but not specifically link with advanced age, that's the reason why we did not writer about this topic
Reviewer 2 Report
The main goal of this review is to analyze age-related changes in different stations involved in pain processing. While the topic is of great interest, the structure of the manuscript and the writing style do not allow to appreciate the huge work done. My goal in this review is to provide you some feedback to understand why this paper is pretty weak at this stage with general and specific comments.
I sincerely encourage the author to entirely rewrite the paper in accordance with coding standards.
General comments
The first impression may be the right one. When I open the pdf of your manuscript I directly showed that this paper will be hard to read. Why? The simple fact is that you abused the line breaks which suggest that the ideas are not developed in a substantial paragraph. Please, read the reference that you used through your manuscript and I am sure that you never find any article with 10 paragraphs in the introduction section and 5 paragraphs in the conclusion section. Rather than juxtapose sentence and idea, please tell us a story.
As expected, some sentences are provided between two others and sometime there is no link between them. In addition, the logic of the story is not clear. We switch from fundamental research with general information about aging and come back to physiological indicator. It is not clear in the introduction part that you will provide some information from animal and human models. You have to delineate you plan in the last paragraph of the introduction section.
Pain is a complex world. The nature of the pain could largely influence the patient pathway. It is clear that nociceptive and neuropathic pain could not be considered as the same pain, although the result is an increase of pain intensity. In the same token, experimental pain could not be considered similarly if it’s induced by heat, capsaicin, pressure, etc. You should clarify all these aspect to provide a clear manuscript.
Specific comments
Line 38-42: While authors indicated 50% of older than 65 suffering from chronic pain in a sentence, they also indicated a 38.5% in the next sentence. One sentence combining these information with a ranging percentage could avoid any confusion.
Line 46-47: There is too much sentence alone. For instance, "Advanced age is a defined risk factor for several painful conditions with a neuropathic component such as postherpetic neuralgia" was not linked with the other. The authors should write their introduction with a continuum process, to develop a story rather than a juxtaposition of information. In this state, the introdcution section is very hard to read.
Line 50-52: please provide details, number of study, number of participants, the mean age of the participants, etc. This comment is applicable to all the manuscript.
Line 53-57: It's not clear why the author switch to a mechanism approach and go back general aspect of aging "presbyopia" "presbyacusis" etc.
Line 90: The author switch from part 1 to 3. please indicate in your introdcution that you will first focus on animal model studies.
Line 111: please used ‘author et al.’ when several authors wrote the paper.
Line 113: why a trademark symbol on the name of the fiber?
Line 113-115: how they did that? please add some details to guide the reader through your manuscript.
Line 120-121: animal or human model?
Author Response
Dear reviewer
Thank you for your comments, we tried to improve the paper based on your suggestions.
We answered your comments in the list below
Line 38-42: While authors indicated 50% of older than 65 suffering from chronic pain in a sentence, they also indicated a 38.5% in the next sentence. One sentence combining these information with a ranging percentage could avoid any confusion. We rewrote the sentence including these informations
Line 46-47: There is too much sentence alone. For instance, "Advanced age is a defined risk factor for several painful conditions with a neuropathic component such as postherpetic neuralgia" was not linked with the other. The authors should write their introduction with a continuum process, to develop a story rather than a juxtaposition of information. In this state, the introduction section is very hard to read.We tried to review the introduction to facilitate reading and improve the link between different informations.
Line 50-52: please provide details, number of study, number of participants, the mean age of the participants, etc. This comment is applicable to all the manuscript.We added the details of the review .
Line 53-57: It's not clear why the author switch to a mechanism approach and go back general aspect of aging "presbyopia" "presbyacusis" etc.We deleted this sentence to avoid confusion
Line 90: The author switch from part 1 to 3. please indicate in your introdcution that you will first focus on animal model studies.We specified this
Line 111: please used ‘author et al.’ when several authors wrote the paper. We corrected the citations
Line 113: why a trademark symbol on the name of the fiber?We corrected the misspell.
Line 113-115: how they did that? please add some details to guide the reader through your manuscript.We added the research details
Line 120-121: animal or human model?We clarified in the text
Reviewer 3 Report
In the Manuscript titled “Chronic Pain In The Elderly: Mechanisms and Distinctive Features”, Authors reviewed the scientific literature, gathering specific pathophysiologic characteristics of pain in the elderly that condition and compromise successful management.
The subject matter is very interesting. The analysis of the mechanisms underlying the difference in pain mechanisms in ageing may pave the way to new therapeutic strategies. Being a narrative review, the result is well balanced and well documented, combining both preclinical and clinical results, thus opening to a wide audience.
I have a single suggestion.
One of the main statements in Conclusion section concerns the contribution of neuroinflammation, described here as preponderant. It follows the suggestion that compounds such as Palmitoylethanolamide may be useful in the management of chronic pain in the elderly. However, this aspect has not been sufficiently described in the text.
I suggest describing separately, in a dedicated paragraph, the contribution of neuroinflammation and the role of N-Acyl- ethanolamines, as well as the indirect action on the endocannabinoid system.
I would like to suggest some sources:
Skaper SD, Facci L, Barbierato M, Zusso M, Bruschetta G, Impellizzeri D, Cuzzocrea S, Giusti P. N-Palmitoylethanolamine and Neuroinflammation: a Novel Therapeutic Strategy of Resolution. Mol Neurobiol. 2015 Oct;52(2):1034-42. doi: 10.1007/s12035-015-9253-8. Epub 2015 Jun 9. PMID: 26055231.
Author Response
Dear Reviewer
Thank you for your note and appreciation.
We expanded the section on microglia activation and neuroinflammation and added the source you mentioned. We did not write an entire paragraph on this topic because the subject is really interesting with lots of research and probably deserves a dedicated paper.
Round 2
Reviewer 2 Report
-
Author Response
Dear Reviewer
We modified the layout reducing the number of paragraphs, corrected misspelled words and tried to improvised the readability.